# A modified method to analyse cell proliferation using EdU labelling in large insect brains

Amaia Alcalde Anton[ID]*, Max S. Farnworth[ID]©, Laura Hebberecht©, C. Jill Harrison, Stephen H. Montgomery[ID]*

School of Biological Sciences, University of Bristol, Bristol, United Kingdom

© These authors contributed equally to this work.
* a.alcalde@bristol.ac.uk (AAA); s.montgomery@bristol.ac.uk (SHM)

**Data Availability Statement:** This is a lab protocol which does not report data.

**Funding:** This work was funded by an ERC Starter Grant (758508) and a NERC IRF (NE/N014936/1)

## Abstract

The study of neurogenesis is critical to understanding of the evolution of nervous systems. Within invertebrates, this process has been extensively studied in *Drosophila melanogaster*, which is the predominant model thanks to the availability of advanced genetic tools. However, insect nervous systems are extremely diverse, and by studying a range of taxa we can gain additional information about how nervous systems and their development evolve. One example of the high diversity of insect nervous system diversity is provided by the mushroom bodies. Mushroom bodies have critical roles in learning and memory and vary dramatically across species in relative size and the type(s) of sensory information they process. Heliconiini butterflies provide a useful snapshot of this diversity within a closely related clade. Within Heliconiini, the genus *Heliconius* contains species where mushroom bodies are 3–4 times larger than other closely related genera, relative to the rest of the brain. This variation in size is largely explained by increases in the number of Kenyon cells, the intrinsic neurons which form the mushroom body. Hence, variation in mushroom body size is the product of changes in cell proliferation during Kenyon cell neurogenesis. Studying this variation requires adapting labelling techniques for use in less commonly studied organisms, as methods developed for common laboratory insects often do not work. Here, we present a modified protocol for EdU staining to examine neurogenesis in large-brained insects, using Heliconiini butterflies as our primary case, but also demonstrating applicability to cockroaches, another large-brained insect.

## 1. Introduction

The study of neurogenesis is critical to understanding the evolution and development of nervous systems. Neurogenesis is the process by which neural progenitor cells (neuroblasts) divide and ultimately generate neurons, a process with common features across vertebrates and invertebrates [1]. In invertebrates, this process has been most intensely studied in the fruit fly, *Drosophila melanogaster* [2–5], owing to the exhaustive availability of genetic tools, but there are also isolated but key insights provided by other insect species that reveal conserved and divergent features of brain development [6–8]. For example, consistently across insect species

to SHM. The funders had no role in study design, data collection and analysis, decision to publish, or preparation of the manuscript.

**Competing interests:** The authors have declared that no competing interests exist.

[7, 9–12], so-called type I neuroblasts (NB), divide asymmetrically multiple times, generating one ganglion mother cell (GMC) with each division, and self-renewing for the next cycle. GMCs subsequently produce two identical cells, either neurons or glial cells. Adult cell number is therefore largely determined by the number of rounds of neuroblast division. Type II neuroblasts, which account for a relatively small number of cell lineages in *Drosophila* [13], instead divide symmetrically to generate another cell type, intermediate progenitor cells (IPCs). These IPCs subsequently divide asymmetrically leading to self-renewal and a GMC, which terminally divides symmetrically producing two neurons. Adult cell numbers in these lineages are therefore determined by the number of rounds of neuroblast and IPC division. Hence, by adding a second proliferative phase IPCs increase the final number of neurons produced by neuroblasts [14–17].

Nervous systems in insects are extremely diverse in their size, structure and ontogenetic trajectories [18, 19]. This variation is likely explained by altered dynamics of cell proliferation, including variation in neuroblast number, the length of neurogenic cell division, and the propensity to produce IPCs. For instance, *D. melanogaster* has 4 mushroom body neuroblasts per hemisphere whereas the honey bee, *Apis mellifera* reportedly has 500 mushroom body neuroblasts. Another axis of interspecific variation can be seen in the timing of key neurodevelopmental events [20]. This diversity of insect neurodevelopment offers the possibility of uncovering mechanisms that govern key cellular processes, leading to a better understanding of the development and evolution of nervous systems. However, to exploit this diversity we must overcome the challenge of developing or optimising methodologies for less frequently studied organisms, for which protocols developed in more common lab insects may be ineffective.

One example of insect nervous system diversity is the size of a central brain structure called the mushroom bodies. Mushroom bodies are one of the most prominent and variable structures in the insect brain. They have a variety of functions, including sensory integration, filtering and attention [21–23]. However, they are particularly implicated in learning, memory, and generally complex behaviours that require the integration of innate states and sensory stimuli [19]. Across insects, mushroom bodies have changed in size multiple times, with particularly large mushroom bodies evolving in at least four lineages [24–27] providing opportunities to study the convergent evolution of expanded neural structures. This size variation reflects changes in the number of intrinsic mushroom body neurons, the Kenyon cells [24, 28]. The developmental mechanisms behind these different population sizes remain largely unclear. Studying the rate and duration of neurogenesis in a comparative context across related species with diverse neural morphologies could advance our understanding of the developmental mechanisms controlling cell production.

To explore variation in Kenyon cell production we focus on one of the four noted increases of mushroom body size, which occur in passion vine butterflies, *Heliconius* [26, 29]. Relative to the rest of the brain, *Heliconius* mushroom bodies are 3-4X larger than other genera in the tribe Heliconiini [26, 28, 29]. The close relatedness of the Heliconiini [30], and the general similarity in their ecology [31] and juvenile life history [32], provides a clear opportunity for comparative studies of development. However, this system lacks basic tools to study neurogenesis.

In other insects, three methods have been used to identifying neurogenesis: i) methods based on the incorporation of chemical markers during the S phase or M phase of the cell cycle; ii) methods that use markers against specific proteins expressed in the membrane of proliferating cells; and iiii) genetic tools. The last two groups of methods have primarily been developed for *Drosophila*, with genetic tools currently being less tractable in other systems. We therefore focused on the first group of methods which can also provide information about cell activity.

The first method used to detect proliferating cells in this way was [³H]-thymidine autoradiography [33]. In this technique [³H]-thymidine binds to the DNA of cells undergoing mitosis and labelled DNA is detected by autoradiography [33]. With this method, Altman reported adult neurogenesis in the human dentate gyrus [34]. The main limitations of this technique are the requirement for use of a radioisotope and the time that the detection takes, which can last months [35]. A related method to label cells in mitosis uses 5-Bromo-2-deoxyuridine (BrdU), a synthetic analogue of thymidine, that binds to DNA during the S phase. This method is faster and provides higher temporal and spatial resolution. It has been widely used to mark neuroblasts in the insect brain [36–39] and to study adult neurogenesis in both vertebrates [40] and invertebrates [37, 41], including moths [42]. Apart from being technically difficult, the main limitation, in this case, is that it requires a strong denaturization of the DNA which can degrade the structure of the sample and adversely affect the tissue's morphology.

More recently a similar method has been developed that uses an alternative nucleotide analogue, 5-ethynyl-2-deoxyuridine (EdU) [35]. EdU is similar to BrdU but the nucleotide is detected by a chemical reaction. Compared with BrdU this method is more sensitive, faster and does not require DNA denaturalization which permits better conservation of cellular structure [35]. Additionally, studies comparing EdU and BrdU have indicated that EdU is more effective at detecting cell proliferation and easier to use [43, 44]. Other markers such as the anti-phospho-Histone H3 (Ser10) antibody have been used to mark cells in mitosis. The process of phosphorylation of Histone H3 starts during the G2 phase but gets reversed at the end of mitosis. Therefore, this label serves as a momentary "snapshot" rather than a long-lasting indicator. In comparison with pH3, EdU marks every cell undergoing S-phase, so it captures a wider picture of divisions instead of smaller windows of the cell cycle. It has also been used to study neurogenesis [7], but currently in a restricted range of small insect species which are commonly used as model organisms, *Drosophila melanogaster* and the red flour beetle, *Tribolium castaneum* [7, 45]. To our knowledge, in Lepidoptera and other large insects, only [³H]-thymidine, BrdU, and histological images have been used to identify dividing cells and their progeny [42, 46, 47]. Although these methods have been very useful, they are limited by the time required, lower sensitivity and, in the case of the histological images, the lack of information about cell activity. In this work, we therefore adapt existing EdU staining protocols to study neurogenesis in large-brained insects, using Heliconiini butterflies as a focal case study.

## 2. Material and methods

The protocol described in this peer-reviewed article is published on protocols.io (https://dx. doi.org/10.17504/protocols.io.n92ldmy69l5b/v1) and is included for printing purposes as S1 File." Experimental variations in the protocol and their outcome are summarized in S1 Table in S2 File and EdU incorporation is further explained in this section. S1 Table in S2 File shows the different combinations trialled here, including those which were successful and unsuccessful, to increase efficiency when designing future experiments.

### 2.i Animal husbandry

As a positive control in early experiments we used Oregon R. wild-type flies of *Drosophila melanogaster* (2,500 Kenyon cells/hemisphere [48]) kept in standard laboratory conditions at 25ºC. We selected and collected flies in the prepupal stage for our experiments. We included comparisons between two Heliconiini, the red postman butterfly, *Heliconius erato* (52,000 Kenyon cells/hemisphere [48]) and the flame butterfly, *Dryas iulia* (13,000 Kenyon cells/hemisphere [48]) obtained from breeding stocks established from commercial pupae suppliers (The Entomologist Ltd, East Sussex, UK). Butterflies were maintained in ~2m x 2m x 2m cages at

24ºC– 30ºC and 80% humidity 80%. Each cage contained natural host plants for each species, *Passiflora biflora* and *P. triloba* for *D. iulia*, and *P. biflora* for *H. erato*. Butterflies were fed every other day with a pollen/sugar solution (5% pollen or artificial amino acids source, 20% sugar, 75% water). Fresh flowers were also provided from *Lantana* and *Psiguria* as additional sources of food. Larvae were reared in individual pots and fed every day with fresh leaves. Young pupae (0–1 days old) were collected to test EdU staining protocols. To provide an additional comparison of whether the protocol developed for Heliconiini worked in other large-brained insects, we obtained 2–4 weeks old Pacific beetle cockroach, *Diploptera punctata*. Cockroaches came from stock populations at the University of Bristol maintained at 26 ºC and 60% humidity.

### 2.ii EdU incorporation

The protocol is a modified version from the instructions of the Click-iT™ EdU Cell Proliferation Kit for Imaging, Alexa Fluor™ 488 dye (Thermofisher, #C10420). The first step of EdU staining protocols is the incorporation of the EdU nucleotide into the DNA of replicating cells. It therefore requires the cells to be alive for the duration of the incubation. Based on previous studies in *Drosophila* and *Tribolium* [7, 45] we diluted EdU in 0.1M PBS in three different concentrations: 10 μM, 20 μM and 50 μM. Pilot experiments in *Drosophila* led to a final dilution of 20 μM for subsequent trials. As an alternative medium, we tested the benefits of replacing the dilution buffer with Grace's Medium (*ThermoFisher*, #11595030) which is used to maintain insect cell cultures [49].

To refine the EdU protocol we focused on larvae and early pupae, where we anticipated high rates of neurogenesis. We tested four main ways of incorporating EdU (Fig 1).

- Injection: We trialled injecting the solution into the thorax of young pupae using a glass capillary and a micromanipulator. We injected a volume of approximately 5–10μl, followed by incubation times of either 6 or 20 hours. After the incubation period brains were dissected out of the head and fixed.

- Coated host plant: For larvae, we trialled covering leaves of *Passiflora biflora* in EdU solution using a paintbrush. We fed these leaves to fifth instar larvae over 24 or 48 hours before dissecting out the brains and fixing them.

- *Ex vivo* incubation: Brains of young pupae were quickly dissected in 0.1M PBS and moved to an Eppendorf tube containing EdU solution diluted in PBS. As a variation of this step, EdU was alternatively diluted in in Grace's medium. Incubation times were varied from 30 minutes to 4 hours.

- *In vivo* incubation: Finally, we trialled incubations in intact pupae by opening a small window in the cuticle of the pupa (as indicated in the Fig 1), and then transferring the pupa upside-down into an Eppendorf tube containing the EdU solution. Incubation times were varied from 30 minutes to 4 hours.

Out of all these conditions, the *ex vivo* incubation of brain in EdU diluted in Grace's Medium and the *in vivo* incubation were the only successful trials. We found injected larvae often displayed a strong immune response, producing large amounts of melanin, which may have inhibited the incorporation of EdU, while food preparations for larvae likely required higher concentrations of EdU which may become costly. *Ex vivo* incubations without Grace's Medium also failed, which we assume reflected the low survival time for cells in extracted brains.

Ways of incorporating EdU

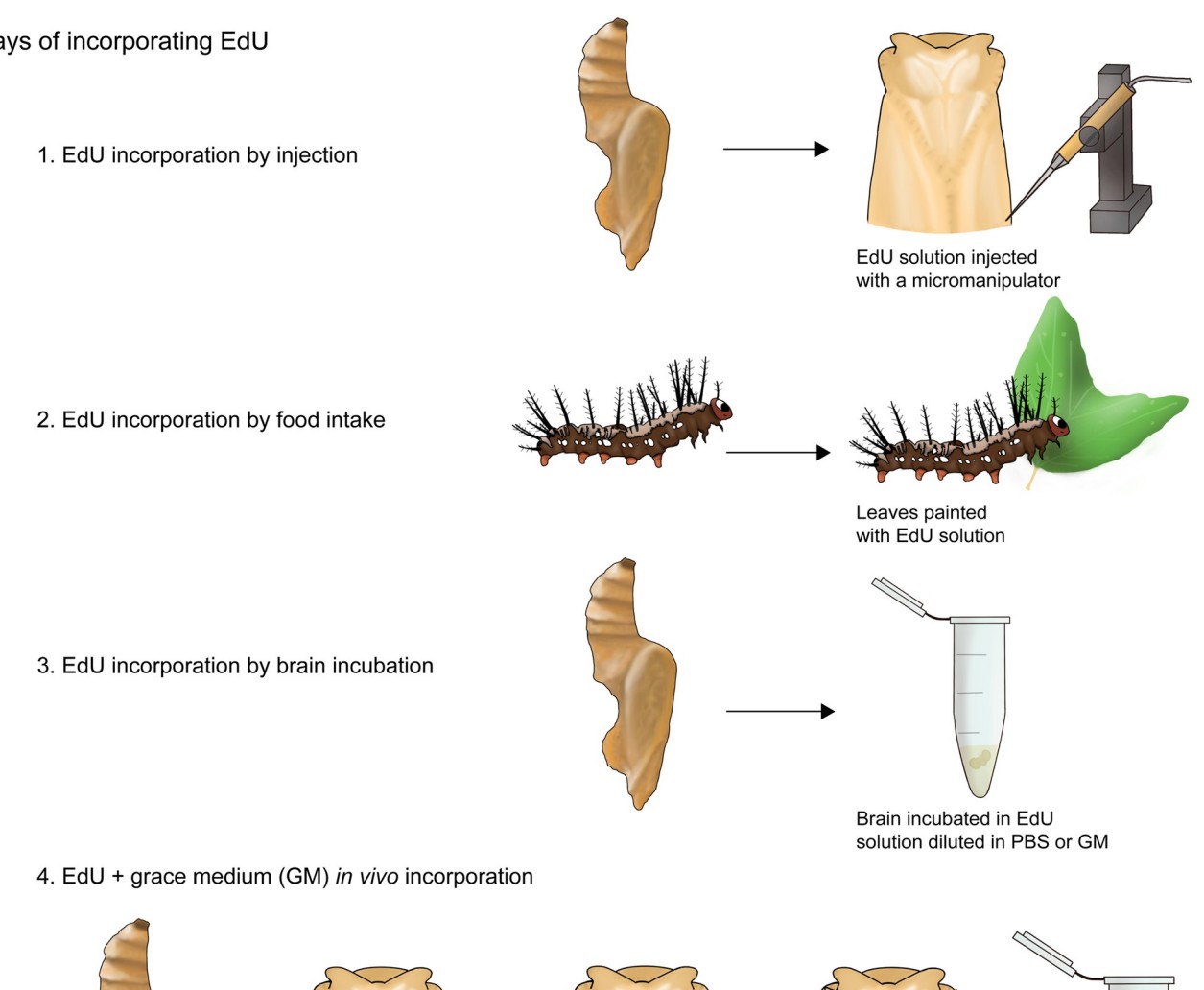

1. EdU incorporation by injection

EdU solution injected
with a micromanipulator

2. EdU incorporation by food intake

Leaves painted
with EdU solution

3. EdU incorporation by brain incubation

Brain incubated in EdU
solution diluted in PBS or GM

4. EdU + grace medium (GM) *in vivo* incorporation

Small window
in the cuticule

Pupa incubated
in EdU + GM

**Fig 1. Different ways of incorporating EdU that were trialed during our protocol development.** From top to bottom: injection into the pupae, painting EdU onto plant material in larvae, and brain and whole pupa incubations.

### 2.iii Immunohistochemistry (IHC)

To illustrate the compatibility of EdU and IHC, we combined our successful protocol derivation with additional antibody staining following the protocol published by Ott [50]. First, under isotonic HEPES-buffered saline (HBS; 150 mM NaCl; 5 mM KCl; 5 mM CaCl$_2$; 25 mM sucrose; 10 mM HEPES; pH 7.4) we dissected a window in the head of a pupa or adult leaving the brain exposed. Heads were then fixed under agitation for 16–20 hours in zinc

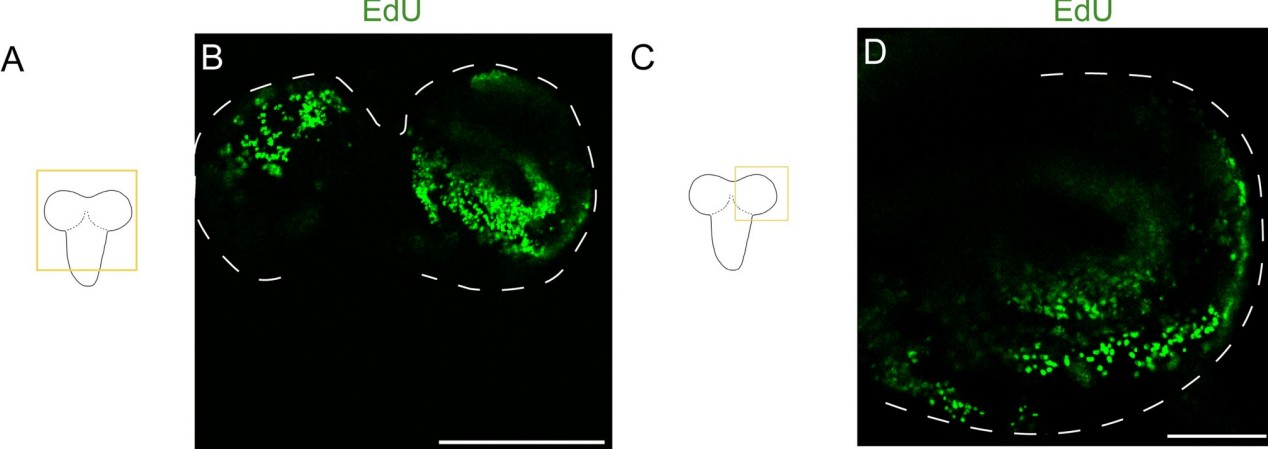

**Fig 2. Positive EdU staining in *Drosophila melanogaster* (prepupa, control).** A and C, Schematic drawings of the brain of the *Drosophila* larvae. B, EdU staining in the central brain. D, EdU staining shown in higher magnification for one lobe. Scale bars = 200 μm in B, 50 μm in D.

formaldehyde, ZnFA (0.25% [18.4 mM] ZnCl2; 0.788% [135 mM] NaCl; 1.2% [35 mM] sucrose; 1% formaldehyde) or 4% paraformaldehyde, PFA, in 0.1 M phosphate-buffered saline (PBS; 7.4 pH). Brains were then dissected out of the head capsule and rinsed in HBS buffer. They were subsequently incubated in PBS with normal goat serum, PBSd–NGS (NGS; 5% Normal Goat Serum; DMSO; 1% dimethyl sulfoxide; 0.005% NaN3 in [0.1 M] PBS), for 2 hours, before incubation with the primary antibody, for 3.5 days at 4°C. As a primary antibody we used a rabbit antibody against horseradish peroxidase HRP, a common marker for neurons [51].

After incubation with the primary antibody, brains were washed three times, 2 hours each time in PBSd (1% DMSO in 0.1 M PBS), before incubation with Cy3-conjugated anti-rabbit antibody 1:100 in PBSd-NGS for 2.5 days. They were clarified using progressively more concentrated glycerol solutions in Tris (1% DMSO in 0.1 M Tris): 1%, 2%, 4% for 2 hours each, 8%, 15%, 30%, 50%, 60%, 70%, and 80% for 1 hour each. They were then dehydrated in 100%

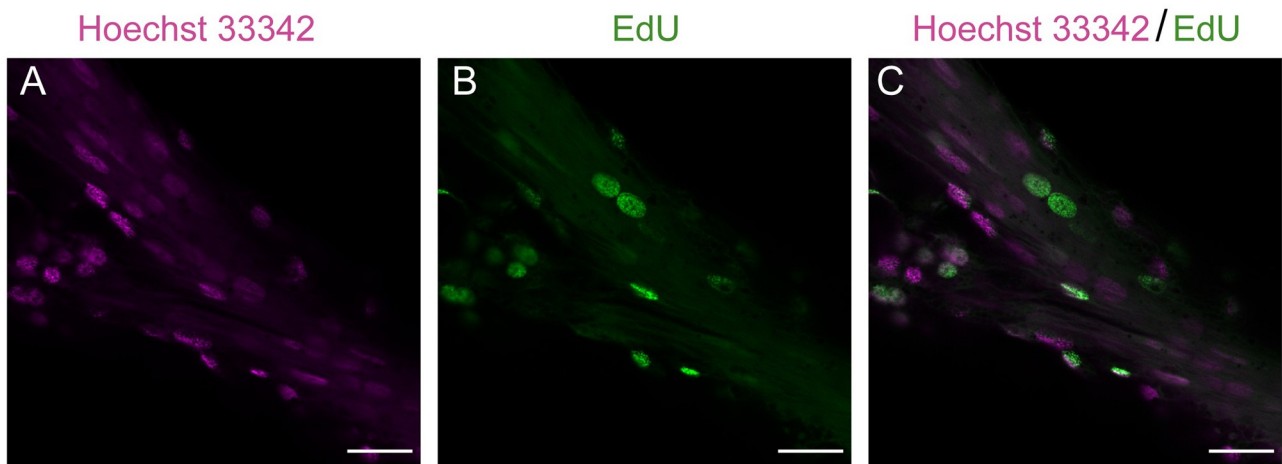

**Fig 3. Nuclei stained by EdU in the optic lobe of *Dryas iulia* young pupa (Day 1).** A, Nuclei staining with Hoechst 3342 (magenta). B, EdU staining (green). C, EdU and nuclei double staining. Scale bars = 20 μm.

ethanol three times, for 30 minutes each, before ethanol was carefully replaced by methyl salicylate letting the brain absorb this solution, becoming transparent in the process.

## 2.iv Confocal imaging and image processing

Butterfly and cockroach brains were mounted in methyl salicylate, brains of *Drosophila* were mounted in 80% glycerol. Brains were imaged using a confocal laser-scanning microscope (Leica TCS SP5, and Leica SP8 AOBS) using 10x HCX PL Fluotar (Numerical aperture: 0.4) and 20x HCX PL APO (Numerical aperture: 0.75) dry objective. For higher magnification images, we used a 40x HC PL APO CS2 (Numerical aperture: 1.3) oil immersion objective. The resolution of the images was 512 x 512 pixels. Images obtained with the confocal were edited using Fiji [52]. Brightness, contrast and colours were adjusted using the colour and channel tools.

To assess EdU penetration we examined the Z-stacks of three different developmental stages and measured the distance to the nearest edge of the brain using Fiji [52]. "Measure" and "set scale" tools were used to calculate the distances. This analysis is included in the S2 Table in S2 File. Based on this analysis we do not detect any penetration issues that might limit downstream analyses.

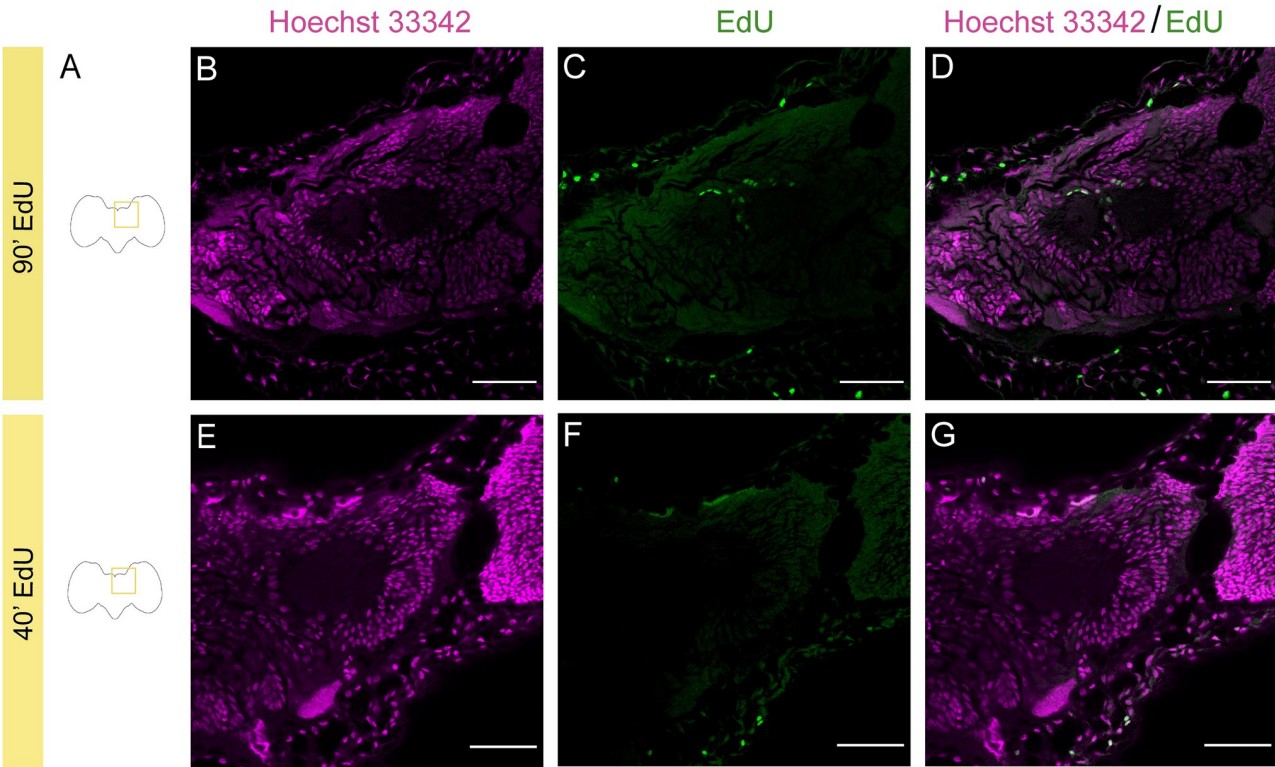

**Fig 4. Positive EdU staining in the mushroom body of *Dryas iulia* young pupae (Day 1), with lower numbers of EdU⁺ cells with shorter incubations.** A, Schematic drawings of the brain of butterflies and location of the staining shown. Nuclei staining with Hoechst 33342 (magenta) after 90' (B) and 40' incubation period (E). EdU staining (green) after a 90' (C) and 40' incubation period (F). EdU and nuclei double staining after a 90' (D) and 40' incubation period (G). Scale bars = 50 μm.

## 3. Expected results

The unedited EdU protocols [35] previously applied in *Drosophila melanogaster* [45] and *Tribolium castaneum* [7] produced good results in our *D. melanogaster* samples (Fig 2), used here as a positive control, but failed to return stained nuclei in Heliconiini. Out of all the experimental variations tested in Heliconiini butterflies (S1 Table in S2 File), the most reliable and effective protocol was *in vivo* incubation in Grace's Medium. The failure of other incubation methods, including those used in *Drosophila* and *Tribolium* highlighted the importance of the EdU incorporation step, and the increased difficulty of this step in larger brained insects. In addition to altering the protocol at the incubation stage, our protocol also includes modifications to the fixation, permeabilization solutions, and incubation times which together adapt the protocol for effective use in larger-brained insects. The main variations of the protocol are summarized in S3 Table in S2 File.

By following the protocol (S1 File) one should be able to obtain similar images to Figs 3–6, which show Edu+ cells in large-brained insects. To ensure that EdU staining was specific to nuclei, we confirmed localised staining at higher magnification (63x) (Fig 3). EdU positive cells were seen to overlap with Hoechst 33342 stained nuclei (Fig 3).

### 3.i Effects of variation in EdU incubation times

Once the method was adapted to butterflies, we focused on the effects of varying incubation times. As EdU marks every daughter cell that was produced during the incubation period

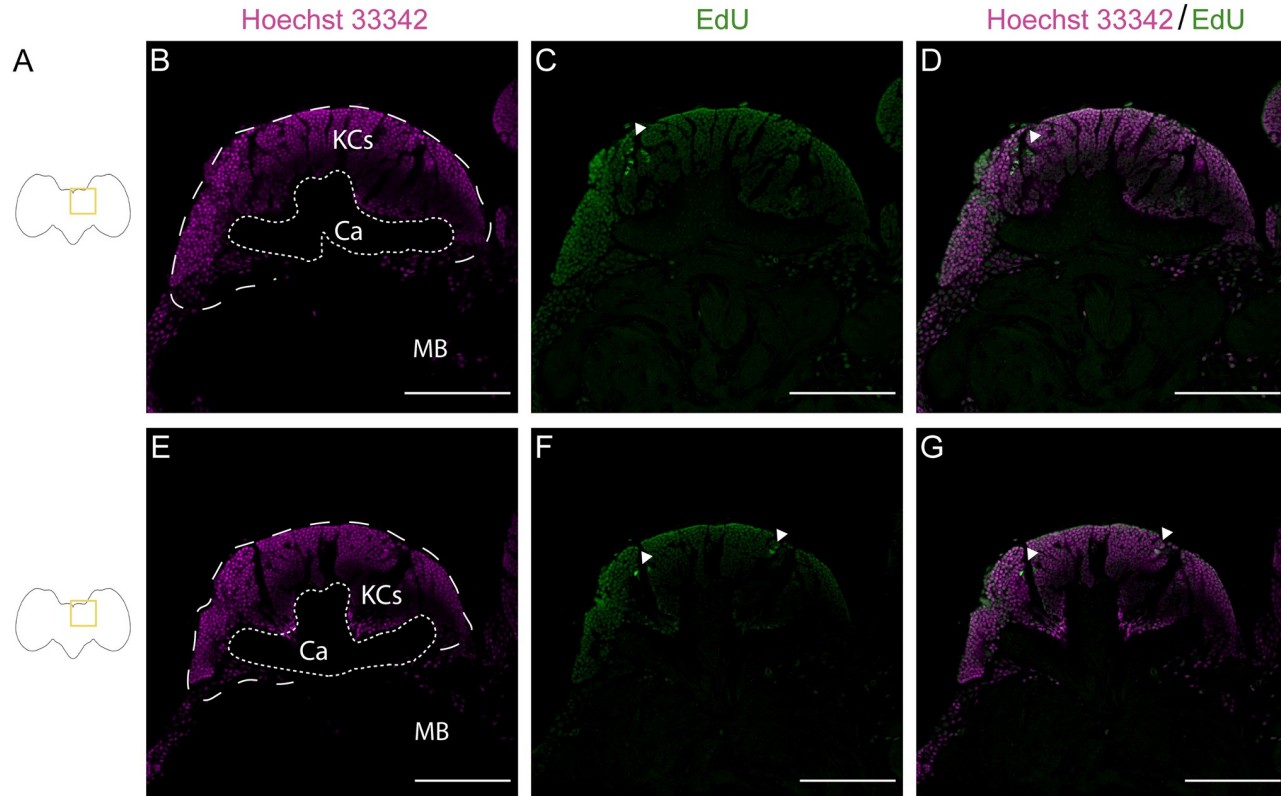

**Fig 5. EdU positive staining in the mushroom body of another large-brained insects, *Diploptera punctata* (4 weeks old).** A, Schematic drawings of the brain showing the location of the imaged cells. B, E, nuclei staining with Hoechst 33342 (magenta). C, F, EdU staining (green). D, G, EdU and nuclei double staining. The arrows indicate EdU⁺ cells. MB: Mushroom body, Ca: Calyx, KCs: Kenyon cells. Scale bars = 100 μm.

(including both self-renewed progenitors and neurons), we expect shorter incubation periods to result in fewer stained cells. In the initial experiment, the incubation with EdU lasted 90 minutes. This period of incubation time was reduced to 40 minutes. As expected, brains stained for 90 minutes show a notably higher number of EdU positive cells compared with brains stained for 40 minutes (Fig 4). Further optimisation of incubation times will likely be required depending on the aims of the particular application.

### 3.ii EdU staining in other large brain insects

To test if the refined protocol developed in butterflies is applicable to other large insects we performed a series of tests in the Pacific beetle cockroach, *Diploptera punctata*. Cockroaches were first dissected in Grace's Medium and later incubated in 20 μM EdU solution diluted in Grace's Medium for 2–3 hours. EdU positive cells were observed among the Kenyon cells (KCs) of 2–4 weeks old cockroaches (Fig 5), suggesting the protocol will likely work across a range of large insect species.

### 3.iii EdU staining in combination with immunohistochemistry

Finally, to demonstrate the compatibility of the altered EdU protocol with immunohistochemistry (IHC) [35], we combined this method with IHC in butterflies. Once the chemical reaction was finished, we washed heads and removed the brains, before proceeding with the antibody staining procedure. Combined staining with anti-HRP, a common neuronal marker [51], was successful, and helps orientate the EdU positive cells and identifying neurons (Fig 6). A full brain hemisphere is shown. EdU+ cells are a mix of progenitor cells and daughter cells, neurons and glia.

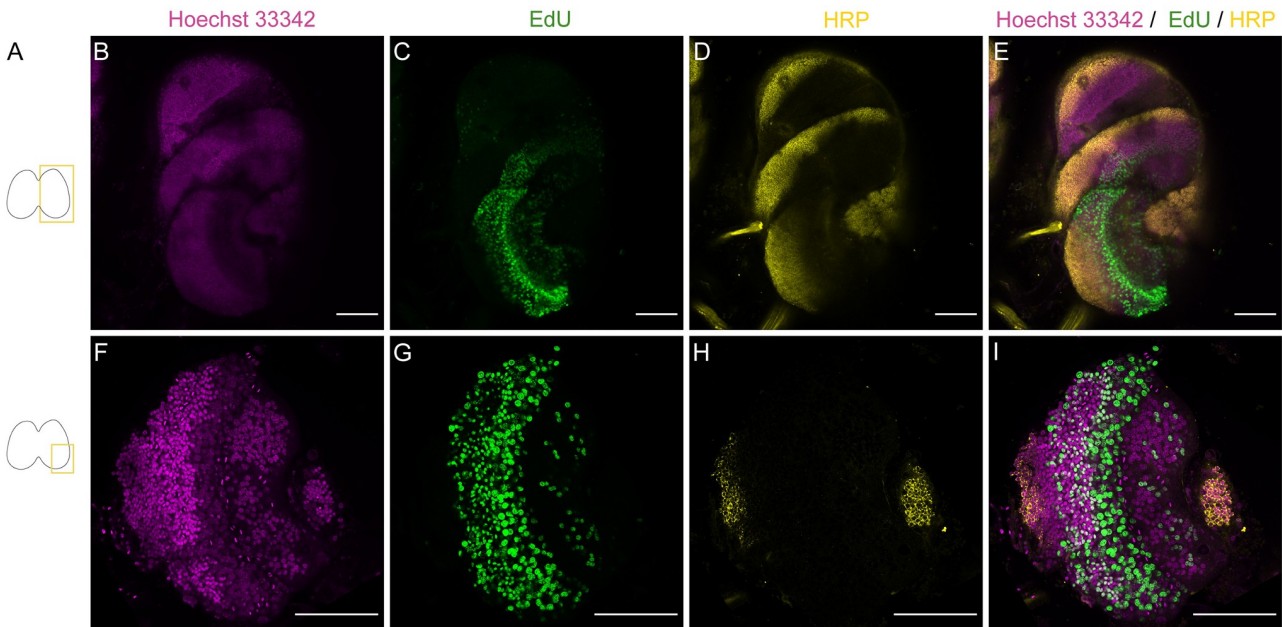

**Fig 6. Compatibility of EdU and immunostainings in one hemisphere of the brain of a *Dryas iulia* larva (5th instar).** Triple staining of Hoechst 33342, EdU and anti-HRP (horseradish peroxidase) in *Dryas iulia* late larva (5th instar). A, Schematic drawings of the brain showing the location of the imaged cells. B, F nuclei staining with Hoechst 33342 (magenta). C, G EdU staining (green). D, H, HRP staining (yellow). E, I, triple staining. Scale bars = 100 μm in B-E, 50 μm in F-I.

## Supporting information

**S1 File. Step by step protocol, also available on protocols.io.**
(PDF)

**S2 File.**
(XLSX)

## Acknowledgments

We thank James Chen, Alana Kelly, and Tom Pitman for their assistance with insect and plant husbandry, and Molly Beastall and Sinead English for providing the cockroaches.

## Author Contributions

**Conceptualization:** Amaia Alcalde Anton, C. Jill Harrison, Stephen H. Montgomery.

**Data curation:** Amaia Alcalde Anton.

**Formal analysis:** Amaia Alcalde Anton.

**Funding acquisition:** Stephen H. Montgomery.

**Investigation:** Stephen H. Montgomery.

**Methodology:** Amaia Alcalde Anton, Max S. Farnworth, Laura Hebberecht.

**Supervision:** C. Jill Harrison, Stephen H. Montgomery.

**Validation:** Amaia Alcalde Anton.

**Visualization:** Amaia Alcalde Anton.

**Writing – original draft:** Amaia Alcalde Anton.

**Writing – review & editing:** Amaia Alcalde Anton, Max S. Farnworth, Laura Hebberecht, C. Jill Harrison, Stephen H. Montgomery.

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
