## [Decision Letter · Decision Letter 0]

12 May 2023

PONE-D-23-10274A modified method to analyse cell proliferation using EdU labelling in large insect brainsPLOS ONE

Dear Dr. Alcalde,

Thank you for submitting your manuscript to PLOS ONE. After careful consideration, we feel that it has merit but does not fully meet PLOS ONE’s publication criteria as it currently stands. Therefore, we invite you to submit a revised version of the manuscript that addresses the points raised during the review process.

We look forward to receiving your revised manuscript.

Kind regards,

Wolfgang Blenau

Academic Editor

PLOS ONE

Journal Requirements:

   "This work was funded by an ERC Starter Grant (758508) and a NERC IRF (NE/N014936/1) to SHM." 

   "We thank James Chen, Alana Kelly, and Tom Pitman for their assistance with insect and plant husbandry, and Molly Beastall and Sinead English for providing the cockroaches. This work was funded by an ERC Starter Grant (758508) and a NERC IRF (NE/N014936/1) to SHM."

   "This work was funded by an ERC Starter Grant (758508) and a NERC IRF (NE/N014936/1) to SHM."

4. We note you have not yet provided a protocols.io PDF version of your protocol and/or a protocols.io DOI. When you submit your revision, please provide a PDF version of your protocol as generated by protocols.io (the file will have the protocols.io logo in the upper right corner of the first page) as a Supporting Information file. The filename should be S1_file.pdf, and you should enter “S1 File” into the Description field. Any additional protocols should be numbered S2, S3, and so on. Please also follow the instructions for Supporting Information captions [https://journals.plos.org/plosone/s/supporting-information#loc-captions]. The title in the caption should read: “Step-by-step protocol, also available on protocols.io.”

Please assign your protocol a protocols.io DOI, if you have not already done so, and include the following line in the Materials and Methods section of your manuscript: “The protocol described in this peer-reviewed article is published on protocols.io (https://dx.doi.org/10.17504/protocols.io.[...]) and is included for printing purposes as S1 File.” You should also supply the DOI in the Protocols.io DOI field of the submission form when you submit your revision.

If you have not yet uploaded your protocol to protocols.io, you are invited to use the platform’s protocol entry service [https://www.protocols.io/we-enter-protocols] for doing so, at no charge. Through this service, the team at protocols.io will enter your protocol for you and format it in a way that takes advantage of the platform’s features. When submitting your protocol to the protocol entry service please include the customer code PLOS2022 in the Note field and indicate that your protocol is associated with a PLOS ONE Lab Protocol Submission. You should also include the title and manuscript number of your PLOS ONE submission.

Additional Editor Comments:

It should be explicitly pointed out what special modifications the new protocol has in comparison to the “standard EdU protocol”. The quality of Figures 4 to 6 should be improved (see comments from Reviewer #2). Furthermore, experiments to estimate the tissue penetration of the marker (buzzword: Z-stacks) as well as for autofluorescence in the nervous tissue of insects (buzzword: spectral scan) would be desirable. Finally, the reference list should be revised according to the suggestions of both Reviewers. Further suggestions for improvement can be found in the reports of the two experts (see below).

Reviewers' comments:

Reviewer's Responses to Questions

**Comments to the Author**

1. Does the manuscript report a protocol which is of utility to the research community and adds value to the published literature?

Reviewer #1: Yes

Reviewer #2: No

2. Has the protocol been described in sufficient detail?

To answer this question, please click the link to protocols.io in the Materials and Methods section of the manuscript (if a link has been provided) or consult the step-by-step protocol in the Supporting Information files.

The step-by-step protocol should contain sufficient detail for another researcher to be able to reproduce all experiments and analyses.

Reviewer #1: Partly

Reviewer #2: Yes

3. Does the protocol describe a validated method?

Reviewer #1: Yes

Reviewer #2: No

4. If the manuscript contains new data, have the authors made this data fully available?

Reviewer #1: N/A

Reviewer #2: N/A

**5. Is the article presented in an intelligible fashion and written in standard English?**

Reviewer #1: Yes

Reviewer #2: Yes

6. Review Comments to the Author

Reviewer #1: The authors present a useful protocol for investigating neurogenesis during development of the insect nervous system using the EdU method. This method offers advantages over the main alternative, BrdU-labeling, in that denaturation of the DNA and immunolabeling are not required to identify proliferating cells. It is also likely that small fluorescent azide molecules penetrate tissues better than immunoglobulin molecules, making this method highly suitable for studying wholemount preparations of insect brains, although penetration was not directly assessed in this study and the photomicrographs included could all represent label in the superficial aspects of the tissue. If the authors collected Z-stacks and could estimate tissue penetration of the label, that would be very helpful. I appreciate the care the authors gave to identifying an effective method of delivering the marker to living tissue. My specific comments and questions follow.

1. It might be useful to non-specialists to include the common name of these butterflies (passion-vine or passion flower butterflies) and to note their Neotropical distribution.

2. I appreciate the scholarly overview provided by the citations, but was surprised to see Truman and Bate's 1988 (!) use of BrdU to study Drosophila neurogenesis not acknowledged (Truman JW, Bate M. Spatial and temporal patterns of neurogenesis in the central nervous system of Drosophila melanogaster. Dev Biol. 1988 Jan;125(1):145-57. doi: 10.1016/0012-1606(88)90067-x. PMID: 3119399.). Lovely to see the inclusion of Ruth Nordlander's and John Edwards' studies of monarch butterflies.

3. To assist investigators just entering the field, define technical terms as fully as possible. Instead of "In comparison with other proliferation markers like pH3," make sure the reader knows the reference is to the use of anti-phospho-Histone H3 (Ser10) antibody as a marker for mitosis. Phosphorylation of Histone H3 begins during G2 but is reversed at the end of mitosis, so this label is a "snapshot," not a persisting marker. (Line 122)

4. Along the same lines, help the novice reading line 123 by replacing replication with S-phase.

5. The test mentions both PFA and ZnFA, but the table, if I am reading it correctly, indicates that ZnFA never produced positive results. Can the authors clarify? The provided protocol is based on PFA but the authors mention that ZnFA can also be used. This is confusing.

6. Did the authors perform a spectral scan to select markers that would minimize autofluorescence in insect nervous tissue?

Reviewer #2: This paper describes the adaptation of EdU protocol specifically for big insect species, EdU being very commonly used to label proliferating cells in many animal species including small insects.

Although a very clear description of the protocol is given in supplementary materials, that will be useful for any researcher willing to use this technique, the manuscript suffers from weaknesses:

- The modifications performed compared to standard EdU protocol are not well underlined. It would help to point out in the text what was modified compared to standard EdU protocol to better emphasize the specificities of this new protocol

- It would be interesting to illustrate the result of your protocol on Drosophila to compare the efficacy with standard protocol (identical results, better or worse?)

- Fig4: labeling is blurred, does not look like confocal image (specially for Hoechst)

- Fig 5 (cockroach) please add higher magnification picture

- Fig 6 : the size of EdU labeled spots is too small to be nuclei. Does not look like in Fig3, where EdU clearly labels nuclei. Add Hoechst labeling to clarify

- Discussion: Any idea why EdU incorporation fails in big insect species while BrdU incorporation works (ex in Locusta migratoria, Periplaneta americana)? This first step (injection / incubation) is not different, only the revelation is.

- Introduction : it would be fair to cite the first papers using BrdU to label neuroblasts in insect brain (Cayre et al Nature 1994 et Malaterre et al J Comp Neurol 2002)

- Error in references: ref 50 (Weng M, Komori H, Lee CY. 2013) cited page 7 does not mention the use of EdU

7. PLOS authors have the option to publish the peer review history of their article (what does this mean?). If published, this will include your full peer review and any attached files.

Reviewer #1: No

Reviewer #2: **Yes: **Cayre Myriam

---

## [Author Response · Author response to Decision Letter 0]

10 Jul 2023

Additional Editor Comments:

• It should be explicitly pointed out what special modifications the new protocol has in comparison to the “standard EdU protocol”. The quality of Figures 4 to 6 should be improved (see comments from Reviewer #2). Furthermore, experiments to estimate the tissue penetration of the marker (buzzword: Z-stacks) as well as for autofluorescence in the nervous tissue of insects (buzzword: spectral scan) would be desirable. Finally, the reference list should be revised according to the suggestions of both Reviewers. Further suggestions for improvement can be found in the reports of the two experts (see below).

Many thanks for this summary. We have responded to all these points below which we believe has improved the clarity and quality of the manuscript. 

Reviewer #1: 

• The authors present a useful protocol for investigating neurogenesis during development of the insect nervous system using the EdU method. This method offers advantages over the main alternative, BrdU-labeling, in that denaturation of the DNA and immunolabeling are not required to identify proliferating cells. It is also likely that small fluorescent azide molecules penetrate tissues better than immunoglobulin molecules, making this method highly suitable for studying wholemount preparations of insect brains, although penetration was not directly assessed in this study and the photomicrographs included could all represent label in the superficial aspects of the tissue. If the authors collected Z-stacks and could estimate tissue penetration of the label, that would be very helpful. I appreciate the care the authors gave to identifying an effective method of delivering the marker to living tissue. My specific comments and questions follow.

This is an interesting point. We note that in insect brains the majority of cell bodies are around the outside of the neuropils, so penetration is unlikely to be an issue for most cell types. Indeed, all scans were indeed from whole brains, so penetration has already occurred over reasonable distances. We have checked these images to identify cells within deeper tissue in the central brain. We now include a comment on this in the discussion (lines 249-253) and a table (S2 Table) showing the penetration of EdU staining at different stages of development.

• It might be useful to non-specialists to include the common name of these butterflies (passion-vine or passion flower butterflies) and to note their Neotropical distribution.

We have added this to lines 95, and 158-159. We have also added the common names of the other species mentioned: fruit fly, line (line 50), honey bee, (line 70) and red flour beetle (137).

• 2. I appreciate the scholarly overview provided by the citations, but was surprised to see Truman and Bate's 1988 (!) use of BrdU to study Drosophila neurogenesis not acknowledged (Truman JW, Bate M. Spatial and temporal patterns of neurogenesis in the central nervous system of Drosophila melanogaster. Dev Biol. 1988 Jan;125(1):145-57. doi: 10.1016/0012-1606(88)90067-x. PMID: 3119399.). Lovely to see the inclusion of Ruth Nordlander's and John Edwards' studies of monarch butterflies.

Many thanks for pointing out this omission, we have added this reference to the line119.

• 3. To assist investigators just entering the field, define technical terms as fully as possible. Instead of "In comparison with other proliferation markers like pH3," make sure the reader knows the reference is to the use of anti-phospho-Histone H3 (Ser10) antibody as a marker for mitosis. Phosphorylation of Histone H3 begins during G2 but is reversed at the end of mitosis, so this label is a "snapshot," not a persisting marker. (Line 122)

This is an important point, we have taken care to correct this and other instances.

• 4. Along the same lines, help the novice reading line 123 by replacing replication with S-phase.

Done.

• 5. The test mentions both PFA and ZnFA, but the table, if I am reading it correctly, indicates that ZnFA never produced positive results. Can the authors clarify? The provided protocol is based on PFA but the authors mention that ZnFA can also be used. This is confusing.

Thanks for spotting this, we have clarified this in the table. We have had positive results with ZnFA and PFA. 

• 6. Did the authors perform a spectral scan to select markers that would minimize autofluorescence in insect nervous tissue?

There is a degree of autofluorescence in our images, insect neuropils are known to autofluorescence around 480nm, and the Edu fluorescence was at 488nm. However, the autofluorescence does not appear in the cell nuclei, so this does not interfere with the use of Edu to label nuclei.

Reviewer #2:

• 1. This paper describes the adaptation of EdU protocol specifically for big insect species, EdU being very commonly used to label proliferating cells in many animal species including small insects. Although a very clear description of the protocol is given in supplementary materials, that will be useful for any researcher willing to use this technique, the manuscript suffers from weaknesses: 

We appreciate this positive comment. Indeed, we hope it will save other researchers working on large insects from unnecessarily investing time repeating the lengthy protocol optimisation that we had to perform. Indeed, we invested ~8 months trying alternative approaches before finalising the current approach. 

• 2. The modifications performed compared to standard EdU protocol are not well underlined. It would help to point out in the text what was modified compared to standard EdU protocol to better emphasize the specificities of this new protocol

Thank you for raising this, we have added a table (Table S3) introducing the main variations of this protocol and comparing the main steps with the manufacturer’s protocol and the protocol used in the Drosophila literature (e.g. Daul et al., 2010). The most important changes involve the incorporation of EdU, the fixation and permeabilization solutions, and the incubation times. The table is introduced in lines 245-246. 

• 3. It would be interesting to illustrate the result of your protocol on Drosophila to compare the efficacy with standard protocol (identical results, better or worse?)

We appreciate this point, but we are not aiming to present a more efficient protocol for all insects including Drosophila. Indeed, the Drosophila protocol seem to work well for that species and is more efficient in some regards, so it would seem unnecessary for Drosophila researchers to use our protocol. Several steps are also quite different because of the size of the tissue and the nature of the dissections, and this means a direct comparison with efficiency in Drosophila and large insects would be difficult to make. Our main goal is instead to present a working protocol for large insects, in other words in increase the breadth of efficacy rather than efficiency.

• 4. Fig4: labeling is blurred, does not look like confocal image (specially for Hoechst)

Apologies, the images are somewhat pixelated due to insufficient resolution. We have replaced these images in a revised figure, and done the same for figure 3. We have also increased the magnification so the differences are clearer. 

• 5. Fig 5 (cockroach) please add higher magnification picture

Our intention was to show the overall cell cluster and the position of labelled cells within it. We therefore think the current figure meets this purpose. We have nevertheless also added higher resolution images of the nuclei as insets.

• 6. Fig 6 : the size of EdU labeled spots is too small to be nuclei. Does not look like in Fig3, where EdU clearly labels nuclei. Add Hoechst labeling to clarify

Thank you, we have imaged other brains stained with Hoechst, EdU and HRP were the nuclei are marked more clear and revised the figure accordingly. We also include a higher resolution image.

• 7. Discussion: Any idea why EdU incorporation fails in big insect species while BrdU incorporation works (ex in Locusta migratoria, Periplaneta americana)? This first step (injection / incubation) is not different, only the revelation is.

We note both do work, but with Edu the challenge has primarily been to increase uptake of the relevant chemicals. We are not sure why the success of the delivery differs between the methods. BrdU involves a lot of denaturisation steps which the Edu protocol does not and perhaps it is possible these steps help with the penetration of the secondary antibody. But if the reviewer has any suggestions to explain this difference we would be happy to discuss these in the manuscript. 

• 8. Introduction : it would be fair to cite the first papers using BrdU to label neuroblasts in insect brain (Cayre et al Nature 1994 et Malaterre et al J Comp Neurol 2002)

Indeed it would, apologies for that oversight. We have added these references on lines 119 and 120.

• 9. Error in references: ref 50 (Weng M, Komori H, Lee CY. 2013) cited page 7 does not mention the use of EdU

Corrected, many thanks.

---

## [Decision Letter · Decision Letter 1]

20 Jul 2023

PONE-D-23-10274R1A modified method to analyse cell proliferation using EdU labelling in large insect brainsPLOS ONE

Dear Dr. Alcalde,

Thank you for submitting your manuscript to PLOS ONE. After careful consideration, we feel that it has merit but does not fully meet PLOS ONE’s publication criteria as it currently stands. Therefore, we invite you to submit a revised version of the manuscript that addresses the points raised during the review process.

Reviewer #2 asks for information about the labelled cell types. Appropriate markings could be added to the illustrations. Furthermore, some specific explanations would be helpful for the non-specialist reader, e.g. what exactly is labelled by HRP and why the number of labelled cells is so different in the different images. If you respond adequately to these suggestions, I can probably accept the manuscript without inviting Reviewers again.

We look forward to receiving your revised manuscript.

Kind regards,

Wolfgang Blenau

Academic Editor

PLOS ONE

Journal Requirements:

Reviewers' comments:

Reviewer's Responses to Questions

**Comments to the Author**

1. Does the manuscript report a protocol which is of utility to the research community and adds value to the published literature?

Reviewer #1: Yes

Reviewer #2: No

2. Has the protocol been described in sufficient detail?

To answer this question, please click the link to protocols.io in the Materials and Methods section of the manuscript (if a link has been provided) or consult the step-by-step protocol in the Supporting Information files.

The step-by-step protocol should contain sufficient detail for another researcher to be able to reproduce all experiments and analyses.

Reviewer #1: Yes

Reviewer #2: Yes

3. Does the protocol describe a validated method?

Reviewer #1: Yes

Reviewer #2: Yes

4. If the manuscript contains new data, have the authors made this data fully available?

Reviewer #1: N/A

Reviewer #2: N/A

**5. Is the article presented in an intelligible fashion and written in standard English?**

Reviewer #1: Yes

Reviewer #2: Yes

6. Review Comments to the Author

Reviewer #1: I enjoyed reading this manuscript a second time. I did not re-read with the intent to check every line, but I did note that there is a typographical error on line 289 (minutess instead of minutes).

Neuroanatomical studies on non-model organisms can be very challenging. I appreciate very much these investigators' careful work. I believe other research groups will find these methods useful.

Reviewer #2: Thank you for providing better picture quality in this revised manuscript.

It is quite difficult for the reader to appreciate labeling efficiency since no explanations are provided concerning labeled cells: what are these labeled cells in which brain structure (mushroom bodies are mentioned in the introduction bot obviously cells labeled in Fig.6 are not Kenyon cells)? Why so few cells in figs 4,5 and som many in fig. 6? What does HRP label? Such information would help make sense of what is shown.

7. PLOS authors have the option to publish the peer review history of their article (what does this mean?). If published, this will include your full peer review and any attached files.

Reviewer #1: No

Reviewer #2: No

---

## [Author Response · Author response to Decision Letter 1]

31 Aug 2023

Reviewer #1: 

• I enjoyed reading this manuscript a second time. I did not re-read with the intent to check every line, but I did note that there is a typographical error on line 289 (minutess instead of minutes).

Corrected, Thanks!

• Neuroanatomical studies on non-model organisms can be very challenging. I appreciate very much these investigators' careful work. I believe other research groups will find these methods useful.

Many thanks, that is our intention with this paper.

Reviewer #2: 

• Thank you for providing better picture quality in this revised manuscript.

Thanks for the positive feedback.

• It is quite difficult for the reader to appreciate labeling efficiency since no explanations are provided concerning labeled cells: what are these labeled cells in which brain structure (mushroom bodies are mentioned in the introduction bot obviously cells labeled in Fig.6 are not Kenyon cells)? 

Thank you for your response. Different figures show different brain regions. We have clarified this in the figure legends, lines: 279/fig 3 (the cells shown are in the developing optic lobes), line 295/fig 4 (shows the mushroom body) and lines 325-329/fig 6 shows a complete brain hemisphere of a 5th instar larvae. The main point of this figure is to show that the protocol is compatible with immunohistochemistry. 

To further elaborate, our goal is to validate the effectiveness of the protocol across large-brained insects, regardless of their brain structure. The cell identity is not a major concern as EdU marks every cell that undergoes the S-phase. This will be progenitor cells like neuroblasts, ganglion mother cells and intermediate proliferating cells; and their daughter cells, mainly neurons and glia. The efficiency should not be different in different cell types. The only potential limitation would be if the cell cycle duration exceeded the EdU incubation time, in which case some cell types might remain unmarked. In the present case we illustrate the protocol primarily by focusing on the mushroom body, while demonstrating that it works elsewhere in the brain, in different developmental stages and different insects. Other researchers could use it to mark cells anywhere in the brain of a large insect, or potentially other organs. 

• Why so few cells in figs 4,5 and so many in fig. 6? 

Thanks, that’s a good observation. This is because these figures are showing different developmental stages and different areas of the brain. We want to show that the protocol works in pupae but also in larvae. In figure 4, we are showing the mushroom body of a young butterfly pupa. We are here shortening the incubation, and this results in less cells marked. In contrast, in figure 6, we are showing a late larva (5th instar). During this stage of development, a lot of proliferative cells are dividing and these are detected by EdU. It is a stage where many brain structures are changing/developing. Additionally, we show a full brain hemisphere with focus on the ventral part which still comprises several brain structures. 

In figure 5, we are showing the mushroom body of a 4 week old cockroach, a different species and stage, so potentially fewer cells are proliferating, but this is not the intention of the comparison. We merely intend to show the protocol works in other large species.

• What does HRP label? Such information would help make sense of what is shown.

We indicated this in the methods section “As a primary antibody we used a rabbit antibody against horseradish peroxidase HRP, a common marker for neurons [52]. ”. We have also clarified this on the lines 323-324.

---

## [Decision Letter · Decision Letter 2]

11 Sep 2023

A modified method to analyse cell proliferation using EdU labelling in large insect brains

PONE-D-23-10274R2

Dear Dr. Alcalde,

We’re pleased to inform you that your manuscript has been judged scientifically suitable for publication and will be formally accepted for publication once it meets all outstanding technical requirements.

Kind regards,

Gregg Roman, PhD

Academic Editor

PLOS ONE

Additional Editor Comments (optional):

Reviewers' comments:

Reviewer's Responses to Questions

**Comments to the Author**

1. Does the manuscript report a protocol which is of utility to the research community and adds value to the published literature?

Reviewer #1: Yes

Reviewer #2: Yes

2. Has the protocol been described in sufficient detail?

To answer this question, please click the link to protocols.io in the Materials and Methods section of the manuscript (if a link has been provided) or consult the step-by-step protocol in the Supporting Information files.

The step-by-step protocol should contain sufficient detail for another researcher to be able to reproduce all experiments and analyses.

Reviewer #1: Yes

Reviewer #2: Yes

3. Does the protocol describe a validated method?

Reviewer #1: Yes

Reviewer #2: Yes

4. If the manuscript contains new data, have the authors made this data fully available?

Reviewer #1: N/A

Reviewer #2: N/A

**5. Is the article presented in an intelligible fashion and written in standard English?**

Reviewer #1: Yes

Reviewer #2: Yes

6. Review Comments to the Author

Reviewer #1: Revisions and minor corrections have improved the manuscript, especially the clearer descriptions of the figures for non-experts.

Reviewer #2: Thank you for your answers and for clarifying structures and developmental stages for easier picture interpretation

7. PLOS authors have the option to publish the peer review history of their article (what does this mean?). If published, this will include your full peer review and any attached files.

Reviewer #1: No

Reviewer #2: No

---

## [Editor Report · Acceptance letter]

26 Sep 2023

PONE-D-23-10274R2 

A modified method to analyse cell proliferation using EdU labelling in large insect brains 

Dear Dr. Anton:

I'm pleased to inform you that your manuscript has been deemed suitable for publication in PLOS ONE. Congratulations! Your manuscript is now with our production department. 

Kind regards, 

on behalf of

Dr Gregg Roman 

Academic Editor

PLOS ONE